# Anatomy of Anthropically Controlled Natural Lagoons through Geophysical, Geological, and Remote Sensing Observations: The Valli Di Comacchio (NE Italy) Case Study

**Jarbas Bonetti [1]**, **Fabrizio Del Bianco [2]**, **Leonardo Schippa [3]**, **Alina Polonia [4]**, **Giuseppe Stanghellini [4]**, **Nicola Cenni [5]**, **Stefano Draghetti [6]**, **Francesco Marabini [4]** and **Luca Gasperini [4,\*]**

[1] Department of Oceanography, School of Physical and Mathematical Sciences, Campus Trindade, Federal University of Santa Catarina, Florianopolis 88061-245, Brazil; jarbas.bonetti@ufsc.br

[2] Proambiente Scrl, Bologna, Area della Ricerca CNR, 40129 Bologna, Italy; f.delbianco@consorzioproambiente.it

[3] Dipartimento di Ingegneria, Università di Ferrara, 44122 Ferrara, Italy; schlrd@unife.it

[4] Istituto di Scienze Marine, ISMAR, CNR, 40129 Bologna, Italy; alina.polonia@ismar.cnr.it (A.P.); giuseppe.stanghellini@bo.ismar.cnr.it (G.S.); franco.marabini@ismar.cnr.it (F.M.)

[5] Dipartimento di Geoscienze, Università di Padova, 35131 Padova, Italy; nicola.cenni@unipd.it

[6] Liceo Scientifico "Augusto Righi", 40123 Bologna, Italy; stefano.draghetti@righibo.istruzioneer.it

\* Correspondence: luca.gasperini@ismar.cnr.it

**Abstract:** Newly collected morphobathymetric and seismic reflection data from the Valli di Comacchio coastal lagoons, south of the Po River delta (Northeast Italy), combined with historical, remote sensing, and geodetic data highlight a complex geological evolution during the Holocene, strongly affected by anthropic control. All data allowed us to define the present-day depositional environment of the lagoons and reconstruct their recent (late Pleistocene/Holocene) geo-history. We focused on the effects of the anthropic impacts in modifying the pristine environments created by the Holocene transgression along the Adriatic Sea coast, at the mouth of a major river. They include land reclamation works, artificial damming, channel excavations, fluvial diversions, and a recent (last decades) increase in subsidence rate due to gas and water withdrawals. Despite the development of economic activities, which promoted occupation and exploitation of this area in the last millennia, the post-Glacial evolution of the lagoons shows the important role of inherited morphological features, such as sand ridges and barriers. This complex and relatively well-documented evolution makes the Comacchio lagoons a unique example of deep connections between natural processes and long-term human controls, offering insights into the management policies of these important and delicate environments challenged by global changes.

**Keywords:** Valli di Comacchio; coastal lagoons; anthropic control; subsidence; seismic data; geodetic measures; global changes

## 1. Introduction

As a consequence of their natural and economic importance, and valuable ecosystem goods and services [1,2], coastal lagoons have been managed historically to exploit their resources. Pervasive modifications, including controls of their boundaries, bottom topography, depths, and fresh-to-saltwater exchanges, have been implemented in many lagoons to facilitate, for example, navigation, fishing, aquaculture, and salt mining [3]. However, since these water bodies are ephemeral features, usually in a delicate environmental equilibrium [4,5], they require periodical monitoring because prone to ecological crises, often related to anthropogenic impacts. This monitoring can be considered a first and necessary step for effective and sustainable management.

In a scenario of global climatic change [6], coastal lagoons can be threatened particularly by the effects of sea level rise [7–9]. Accordingly, in recent years, the need to

protect and manage with better practices these environments is gaining attention [3,5]. Yet, there is still a lack of knowledge on many aspects of the processes involved in the interaction of lagoon components, which is fundamental in seeking the development of bio-economic models and decision support tools [1]. This inhibits elaboration of vulnerability assessments, adopted extensively in open coasts [10,11].

The study of these environments involves a combination of approaches, related to biology, physical/chemical oceanography, and geology [12]. Considering the physical support given to the development of lagoons, most investigations can benefit, as a starting point, from the knowledge of their geological setting at different levels of accuracy. In particular, important information arises from the study of morphobathymetry [13] and shallow subsurface stratigraphy [14,15]. The variability of the water-sediment interface, controlled by geological and biological factors such as the sediment transport, diagenesis, presence of bio-constructional features, etc., also give important clues of the acting dynamics which may be regulating erosional and depositional processes [16].

If, on the one hand, direct observations of submerged areas, such as lagoons, are normally difficult due to the presence of water, geological surveys of such environments offer advantages over similar studies onshore. First, sedimentation is generally more continuous in time and space, resulting in more reliable chronostratigraphic records and regional correlations [17]. Second, the acquisition of geophysical data supporting geological interpretation is carried out more quickly and at lower costs, allowing dense-spaced data coverage and 3D or pseudo-3D reconstructions [18–21]. Finally, the overall quality of high-resolution subsurface imaging, particularly seismic reflection profiles, is improved by the presence of water, which optimizes the coupling between the seismic sources and the substratum.

Although coastal lagoons could be interesting targets for geological/geophysical studies, such investigations are challenged by the extreme shallowness of these environments, which does not allow in most cases for safe navigation. This is the main reason for the development of Autonomous Surface Vehicles (ASV), which could perform geophysical surveys in ultra-shallow waters with a low environmental impact and at a fraction of the costs of conventional methods [22].

The focus of this paper is the post-Glacial geological history of the Valli di Comacchio coastal lagoons (Figure 1), in the Po River delta region (NE Italy). This complex system of interconnected bodies of brackish water has as an important variable, i.e., a long-lasting anthropic control, which deeply modified the preexisting natural environment since protohistoric times [23]. The creation of artificial embankments and channels, as well as the extensive land reclamation as a consequence of the historical occupation of the territory, has transformed the pristine lagoons into an "anthropically controlled" natural environment [24,25], involving at a more regional scale the entire Po River plain [26,27].

We used geophysical data (morpho-bathymetry and seismic reflection profiles), in conjunction with remote sensing images (satellite and aerial photos and Lidar-based topographic models), geodetic data, and historical maps to reconstruct a post-LGM (Last Glacial Maximum) environmental/geological evolution of this region.

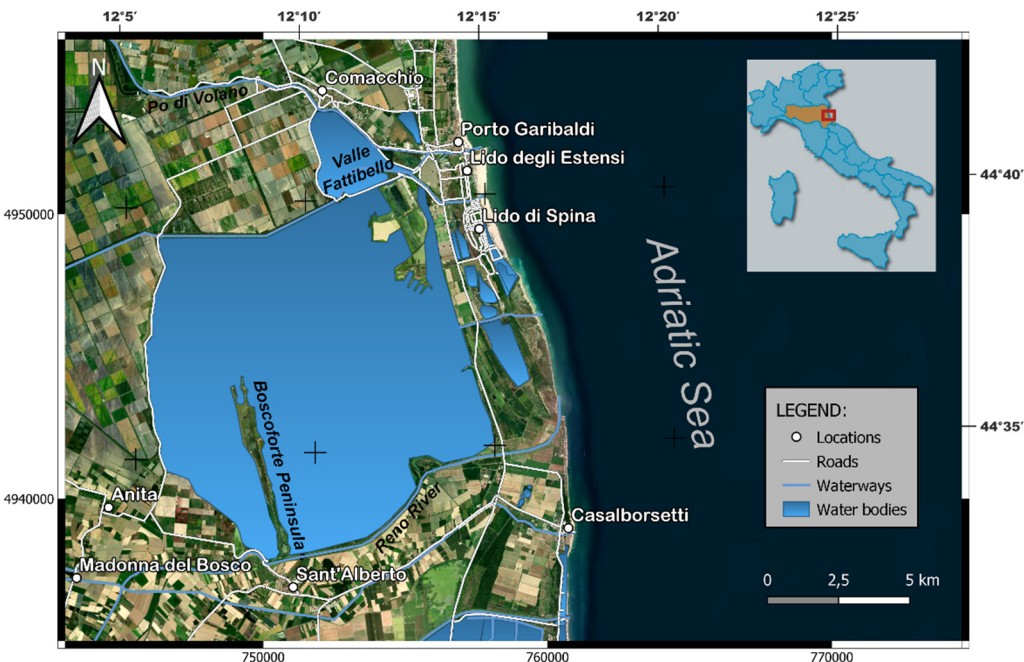

**Figure 1.** The Valli di Comacchio coastal lagoons study site. Coordinates are represented in ED50 99 UTM Zone 32 N (plain characters) and in Geographical W84 (bold characters).

*The Study Area*

Starting from Roman times, agriculture caused a rapid soil loss in the Po River plain due to erosion, which followed deforestation, leading to a significant increase in the sediments supplied to the Adriatic Sea [28–30]. Enhanced sediment supply, together with subsequent natural and artificial diversions of the Po River course, the main by the Venetians in 1604 C.E. (Taglio di Porto Viro), led to the formation of the modern delta, which could be considered a "man-made" feature [31–33].

Land subsidence, caused by human activities and natural processes, has been monitored and investigated by satellite and topographic techniques [34,35]. To understand possible correlations between environmental changes and natural and/or anthropical-induced subsidence, we have analyzed the daily position time-series of GNSS (Global Navigation Satellite System) permanent stations located in the study area, also investigating variations of kinematic subsidence rates.

Although many authors have already studied the Pleistocene/Holocene evolution of the Po River mouth, e.g., [36], there is still a lack of knowledge about morphology and sedimentology of the nearby Valli di Comacchio lagoons, a unique example of a long-standing anthropically controlled transitional environment.

The Valli di Comacchio coastal lagoons are shallow brackish-water environments that extend south of the Po River between the town of Comacchio and the Reno River (Figure 1). These lagoons, named "Valli" formed around the tenth century as a consequence of subsidence processes, and were originally freshwater basins supplied by river floods [37]. Starting from the sixteenth Century, seawater seepages contributed to the progressive formation of the present-day brackish environment, which constitutes the largest wetland remains after massive drainage at the eastern province of Ferrara.

The landscape around the lagoons is flat if we exclude artificial embankments created to limit areas of economic activities and remnants of ancient coastal ridges parallel to the Adriatic coastlines. These zones are important feeding and breeding grounds for migratory and resident birds and have been listed as a "Wetland of International Importance" under the Ramsar Convention since 1981 (https://rsis.ramsar.org/ris/225, 14 February 2022). They were also more recently classified as a regional special protection zone.

Our survey was carried out in the frame of a project for recovery, conservation, and sustainable management of this area, which involved several aspects but started from the knowledge of the "recent" geological evolution of the territory.

Hydrodynamics of the Valli di Comacchio lagoons are controlled by tides, winds, and the inflow of freshwater from several sources. Winds blowing from the west are dominant in frequency but not in intensity, with maximum speeds from the northeast, as observed in a two-year-long series [38]. The lagoons exchange water with the outward waterbodies mainly through several hydraulic structures, including siphons located in the southwest, that take fresh water from the Reno River, and sluice gates operating between the lagoon and the channels, which connect the basin to the Adriatic Sea. The siphons provide a nearly constant man-regulated flow rate, while the flow through the gates is mainly regulated by tidal waves [38]. The tributaries flowing into the lagoons receive water from the Po River and the drainage systems of adjacent reclaimed lands. Several sub-lagoon channels, bounded by partially emerging irregular banks, provide a direct link between the freshwater tributaries and the salt-water channels. The Reno River, which limits the system towards the south, was diverted directly to the Adriatic Sea and does not supply directly the lagoons.

The eastern margin of the Valli di Comacchio is separated from the Adriatic Sea by a sandy barrier of about 2.5 km in width. In a scenario of sea-level changes, with a significant coastal retreat expected for the next decades [6] and increased subsidence due to anthropic activities, this poses a particular threat, since seawater incursion may lead to a different dynamical equilibrium for these lagoons. In fact, in both adjoining open coasts of Ravenna (south of the system) and Ferrara (center and north) provinces, a retreating behavior of the coast with consequent events of coastal flooding and beach erosion has been reported [39,40].

Regional estimates of sea-level changes rates during the Holocene [41] suggest alternating periods of rapid flooding and gradual shoaling stacked in a retrogradational pattern that mostly reflects stepped, post-glacial eustatic rise, and Middle to Late Holocene coastal progradation and delta upbuilding that took place following sea-level stabilization at highstand, starting at about 7 cal ky.

## 2. Materials and Methods

### 2.1. Field Survey

Geophysical data were collected in different periods starting from 2011 and covered the entire accessible water surface of the Valli di Comacchio lagoons, excluding only private areas (Figure 2). Echographic data were collected using a vertical incidence echosounder, the PSA900 manufactured by Datasonics, particularly suitable for shallow-water environments because of its high operating frequency (200 kHz), a narrow (8°, conical) beam width, a short pulse length (350 µs), and a minimum depth range of 0.75 m. However, due to the characteristics of the study area, we modified the echosounder to cover shallower sectors of the lagoons: the pulse length was shortened to 200 µs and the bottom-detection/depth-estimate section was disabled, obtaining, de facto, a 200 kHz externally triggered ultrasonic pinger. In this way, the shallow-depth limit was reduced to approximately 0.2 m. The bathymetric survey lines are displayed in Figure 2. Tidal oscillations were measured during the survey using a tide-recording station consisting of a tide-gauge, a data-logger, and a GNSS receiver, which provided an accurate time-base. The 200 kHz echosounder signal was digitally sampled with a constant time window, and the echograms were stored in SEGY-format files [42]. To reach areas shallower than about 1 m, we used OpenSWAP, an ASV developed in the frame of the NAIADI project [22].

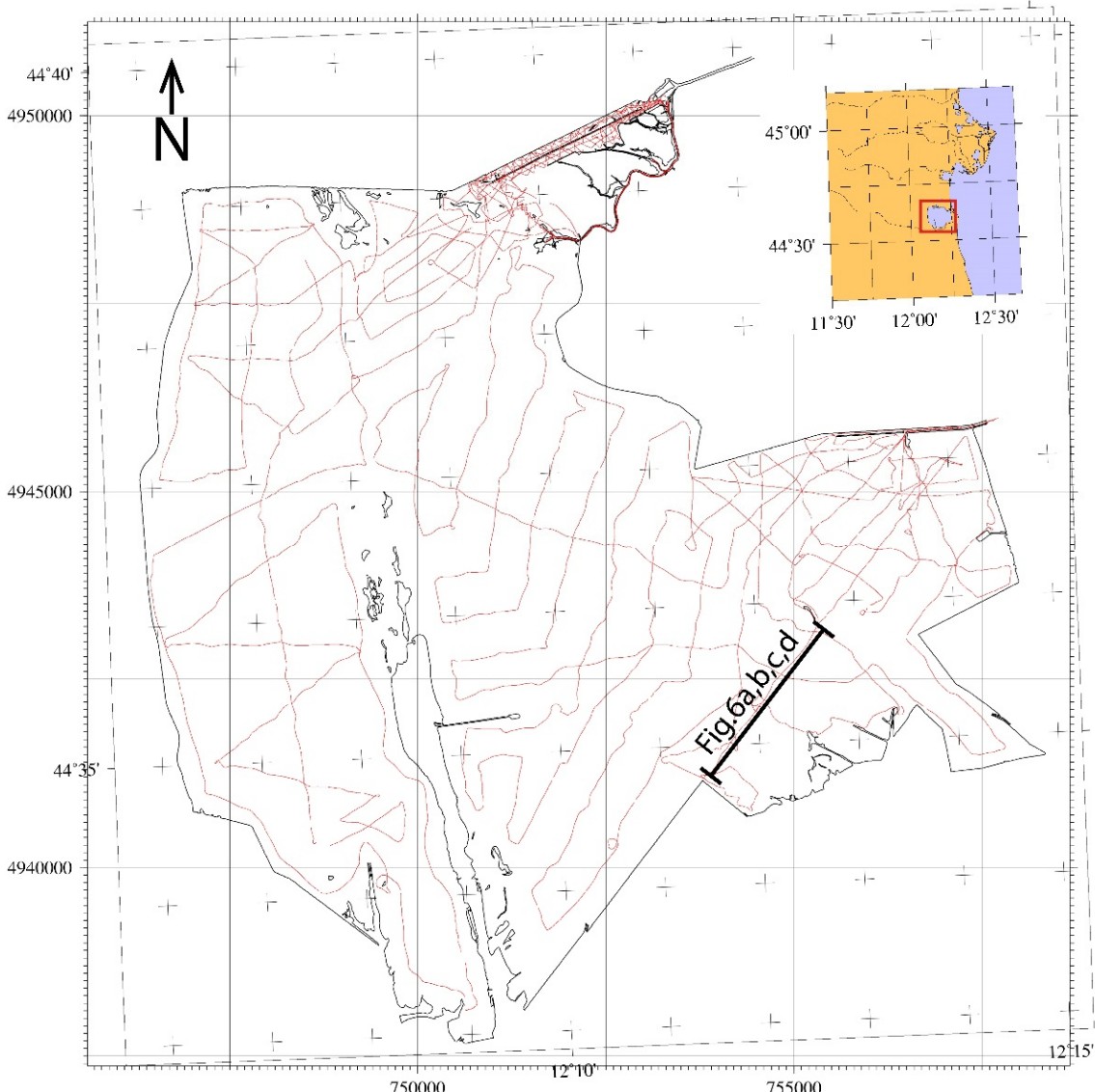

**Figure 2.** Survey lines were performed in the study area (red lines) including bathymetric and seismic reflection data. Projection UTM 32N.

Seismic reflection lines were collected using a Teledyne Benthos Chirp III system, equipped with four transducers mounted on a small catamaran towed by the boat. Data were acquired using a pulse length of 5 ms, frequency from 2 to 7 kHz and sampling rate of 16 kHz. Seismic data were stored in SEGY files in form of instantaneous amplitudes. Vertical resolution of seismic images was about 10 cm and maximum penetration reached 15–20 m.

### 2.2. Data Processing
2.2.1. Morphobathymetry

The first step in data processing was to determine accurately the depths of the water-sediment interface from the raw echograms. The presence of soft sediments and the consequent penetration of the ultrasonic signal below the soft, highly hydrated water/sediment interface affect the accuracy of bottom detection. In order to correct data for this effect, we used the method developed in [43]. Once water-depth measurements were obtained, the tidal correction was performed combining water-depth measurements with the tidal record through the GNSS time-marks. At the end of this process, over 140,000 valid

depth measures were used to compile a netCDF binary grid using a continuous curvature surface algorithm included in the GMT software package, which was also employed to test lines crossings using the x_system [44]. Discrepancies in the water-depth estimate at each intersection (cross-over) of the bathymetric tracks were computed using the program x2sys_cross. Cross-over errors can be large enough to create artificial features in the final gridded dataset, particularly where multiple sources of potential "low-frequency" errors are present, such as tide-effect and GNSS positioning.

### 2.2.2. Bottom Reflectivity

The acquisition of the entire echosounder sweep at each sounding point, rather than the simple depth estimate, gave us the opportunity to calculate the bottom reflectivity. In short, this method allows for the compilation of reflectivity maps, which may be sensitive to some physical/geological properties of the bottom sediments [43,45]. It allows for indirect characterization of the different substrates, mostly due to differences in grain size and/or hardness. Propagation and scattering of high-frequency acoustic sound at or near the bottom is controlled by several factors, including biological, geological, biogeochemical and hydrodynamic processes operating at the benthic boundary layer. However, experimental measurements of compressional wave attenuation suggest that the single most important geotechnical property related to acoustic attenuation is the mean grain size of the insonified sediments [46]. Figure 3 reports the empirical relationship between amount of sand in the lagoon-floor sediments and normalized reflectivity in the Valle Fattibello [43].

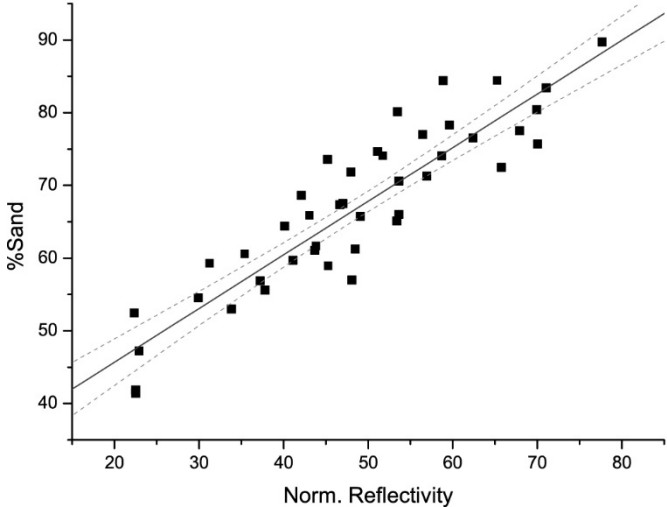

**Figure 3.** Reflectivity vs. % of sand of bottom samples obtained in Valle Fattibello [43]. Reflectivity values were normalized between maximum (71.46%) and minimum (0.85%) occurrence of sand.

### 2.2.3. Time-Slicing

After correcting data by residual statics, we created a "flattened" version of the echograms using a special function of the SeisPrho software. The flattened profiles were obtained by time-shifting the seafloor to a horizontal reference level at 0 ms TWT. In this way, a new set of "flattened" SEGY files was created and subsequently sampled using the Time-Slice function available in SeisPrho, which allows integration of squared seismic amplitudes (proportional to the seismic energy of reflections) within different time windows defined on the basis of local stratigraphy. We used a 1 ms-wide TWT window, which integrates most of the shallower seismostratigraphic unit. This cumulative amplitude index determined for each trace has been considered as an estimate of seismic *facies* character, and an index of lateral coherence of selected reflectors. The same technique has been used to map lateral variations of the substratum seismic response, i.e., the seismic *facies* variability, which could be associated with geological properties [18–20]

### 2.2.4. Differential Bathymetric Models

To assess morphological changes in the lagoon-floor at a decadal time-scale, the newly collected data were compared to a bathymetric survey performed by IDROSER (now ARPA-ER) in 1992. Although having a lower resolution (about 250 m of point spacing), these old soundings presented a good spatial coverage, consistent with the new acquisitions. Analysis of bottom morphology was performed in a GIS using spatial analysis techniques [47]. Both datasets were continuously spatialized using the Triangular Irregular Network (TIN) method. Obtained bathymetric models were then rasterized and subtracted from each other (old bathymetry minus new bathymetry) by applying map algebra in QGIS raster calculator. A color code/contours map was subsequently compiled to better appreciate morphological differences between the two surveys.

### 2.3. Cone Penetration Tests (CPTu)

Seismostratigraphic data were correlated to available stratigraphic information through six CPTu tests, performed in selected stations along seismic reflection profiles [48]. A cone penetration test with pore water pressure measurement, or CPTu, is a static penetration test with water pressure measurement. The main purpose of CPT (CPTU) testing is to identify subsurface conditions: classify soil, detect layers and measure strength, determine deformation characteristics and the permeability of foundation soils. Using empirical relationships [49] applied to measurements of resistance at the tip, lateral friction, friction ratio and interstitial overpressure, information on mechanical parameters that describe the type of terrain crossed were obtained. Measures were carried out every 2 cm during the soundings, for a 16 m of investigation depth at each station [48]. Correlations between seismic reflection profiles and stratigraphic logs were performed using the open software ChirCor [50].

### 2.4. Geodetic Measurements

The present-day vertical and horizontal deformation pattern of the Valli di Comacchio area can be reconstructed by analyzing the GNSS (Global Navigation Satellite System) permanent station data. Position and meta-information about these stations, including date of first acquisition and observation time span, are reported in Table 1.

**Table 1.** GNSS permanent sites used in this study, including: start/end dates of first and last observation (decimal year); number of effective observations (N); observation time-interval (T, decimal years); position, longitude, latitude (decimal degrees), and ellipsoidal height (H, meters).

| Site | Start Date | End Date | N | T(y) | Longitude | Latitude | H(m) |
|------|-----------|----------|-----|-------|-----------|----------|------|
| COCL | 2016.6434 | 2019.8233 | 666 | 3.18 | 12.18852594 | 44.68992146 | 49.42 |
| CODI | 2007.6315 | 2019.6945 | 3916 | 12.06 | 12.11197254 | 44.83667447 | 45.57 |
| GARI | 2009.5466 | 2019.9986 | 3739 | 10.45 | 12.24943596 | 44.67690139 | 47.75 |
| PTO1 | 2010.5575 | 2019.8233 | 2953 | 9.27 | 12.33405291 | 44.95151834 | 49.31 |
| TGPO | 2008.6544 | 2019.9959 | 3721 | 11.34 | 12.22832082 | 45.00305807 | 49.36 |
| RAVE | 2005.3548 | 2019.9986 | 4236 | 14.64 | 12.20029619 | 44.41275804 | 54.81 |
| RAVS | 2007.1082 | 2019.7658 | 3700 | 12.66 | 12.19188083 | 44.40529434 | 51.80 |

Because the employed GNSS sites belong to different networks [34,51–54] they lack homogeneity in observation periods. This could introduce biases in the kinematic field estimates, particularly for the vertical components, which could be affected by high-frequency human-induced signals. We tried to overcome these problems by analyzing two different data sets, i.e., the complete available time series at each station, and those corresponding to time intervals covered by all stations. For the latter, we considered data collected after 2016 when station COCL (Table 1), the nearest to the study area, was active.

Comparison between the subsidence rates estimated using the entire available observation period and those obtained by post-2016 observations were used to discriminate between natural and anthropic effects.

## 3. Results and Discussion

In recent decades, the Po Plain southeastern sector has suffered subsidence rates between 100–200 mm/y due to the superposition of natural and anthropic effects [34,35]. A peak of about 200 mm/y observed around the '50s was considered primarily related to groundwater pumping from shallow and deep aquifers, as well as to increased gas production from Plio-Pleistocene reservoirs [55]. At the end of the 20th century, the reduction of gas-field production and groundwater withdrawals, a consequence of economic crises and changes in the environmental management policies, have reduced the soil-lowering rate. Recent studies [34,51–54] report rates of about 10 mm/y, similar to those estimated prior to the intensive environmental exploitation which followed the Italian economic expansion in the second half of the 20th century [55].

Differences among rates during the entire period and those considering only the post-2016 data are lower than uncertainties associated with vertical and horizontal (north and east) components (Figures 4 and 5), suggesting that subsidence has been stable in the time period analyzed at a rate of <10 mm/y.

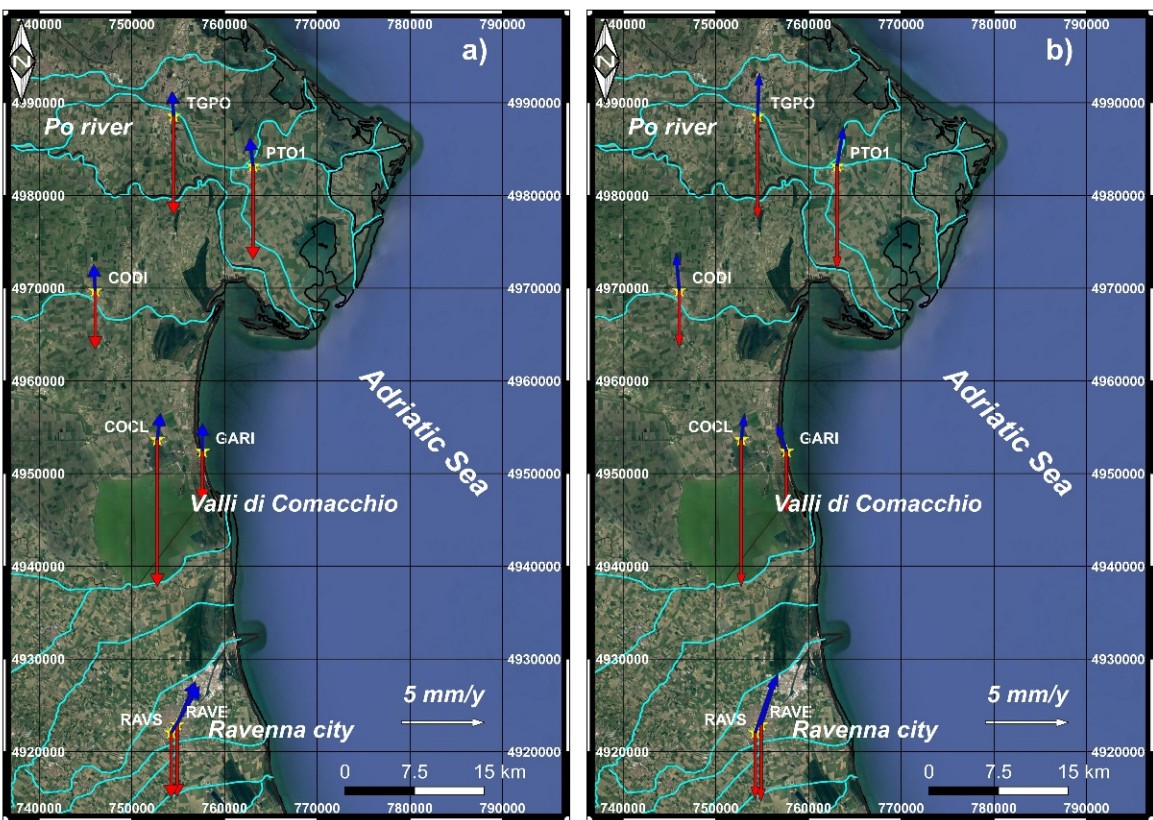

**Figure 4.** Vertical (red arrows) and horizontal (blue arrows) GNSS kinematic patterns in the study area, (**a**) considering the entire time interval, (**b**) data acquired after 2016. Yellow stars indicate positions of the GNSS sites reported in Table 1.

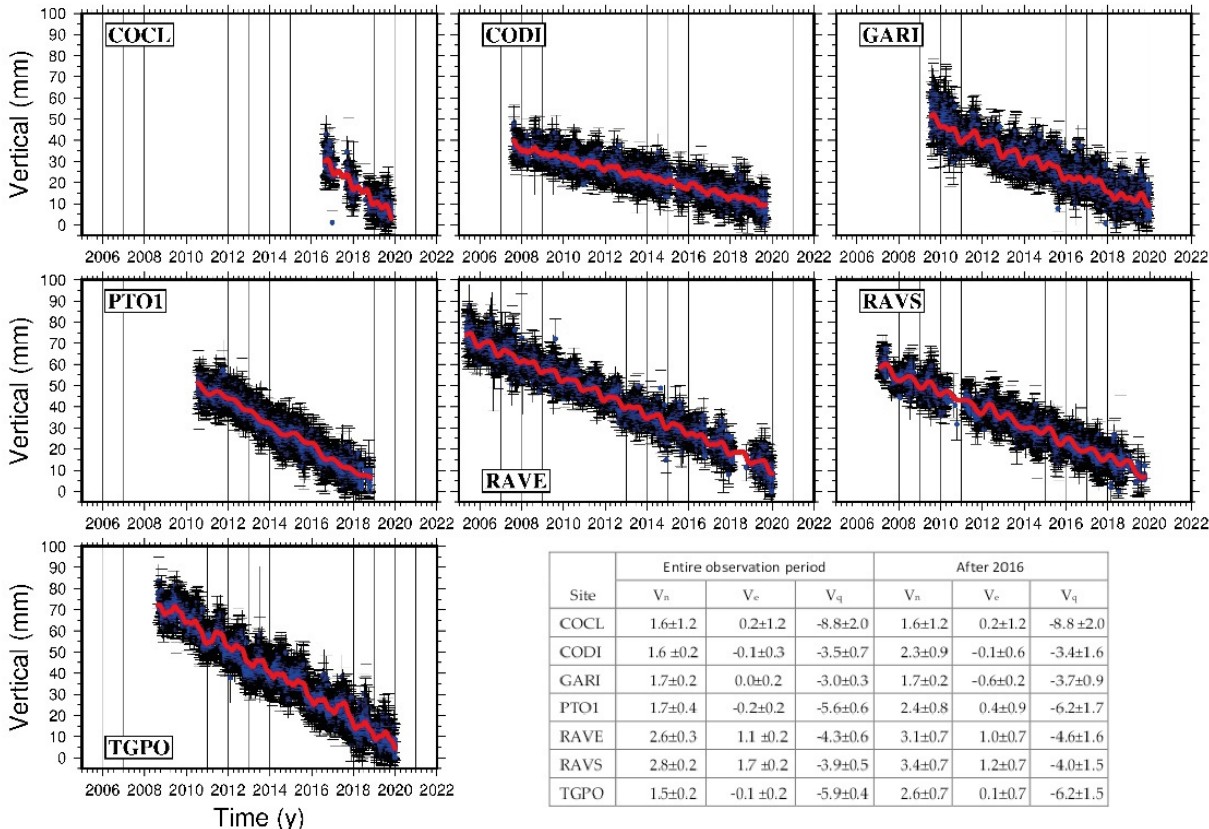

**Figure 5.** Trends of vertical ground motion determined at GNSS stations in the study region (location in Figure 4). Blue dots (and associated black error bars) indicate the vertical component (daily samples); red dots indicate the vertical displacement model estimated using the approach proposed in [34,52–54]. Velocity values adopted to estimate the vertical motion models are reported in the first three columns of the table in the inset, which includes velocities (mm/y) of continuous GNSS stations. The first column reports the CGNSS station codes, V are the velocities in the ETRF2014 reference frame of the north (Vn), east (Ve) and vertical (Vq) components. Uncertainties were estimated using the method proposed in [34,51–54].

## 3.1. Recent Sedimentary Evolution: The Shallow Subsurface

A close-spaced grid of high-resolution echograms and seismic reflection profiles enabled us to study the upper part of the sedimentary sequence that constitutes the substratum of the Valli di Comacchio lagoons down to about 15–20 m of depth. Despite their high resolution, these data suffer the diffuse presence of gas in the sediments, which locally prevented signal penetration. In gas-free areas, the quality of seismic images was enhanced by the peculiar physical character of the water-bottom interface. In fact, the extremely low acoustic impedance contrast at the lagoon-floor generated a weak reflector, allowing a large fraction of acoustic energy to penetrate deeper levels. This occurrence also favored the resolution of acoustic images almost free of multiple reflections.

Lines COM-9 and -10, collected in the SE sector of the lagoon (Figure 6, location in Figure 2), can be considered representative of the shallow stratigraphy of the entire basin. Below the sediment-water interface, we observe fine layered, transparent sediments draping more reflective packages below a marked unconformity (reflector U1, Figure 6), which shows different characteristics within the working area, but, in general, presents a sub-horizontal morphology. The acoustic basement is given by another prominent reflector (U2), which shows, differently from U1, an irregular morphology over the entire area, interpreted as an erosional truncation. Below U2, no coherent reflectors are visible.

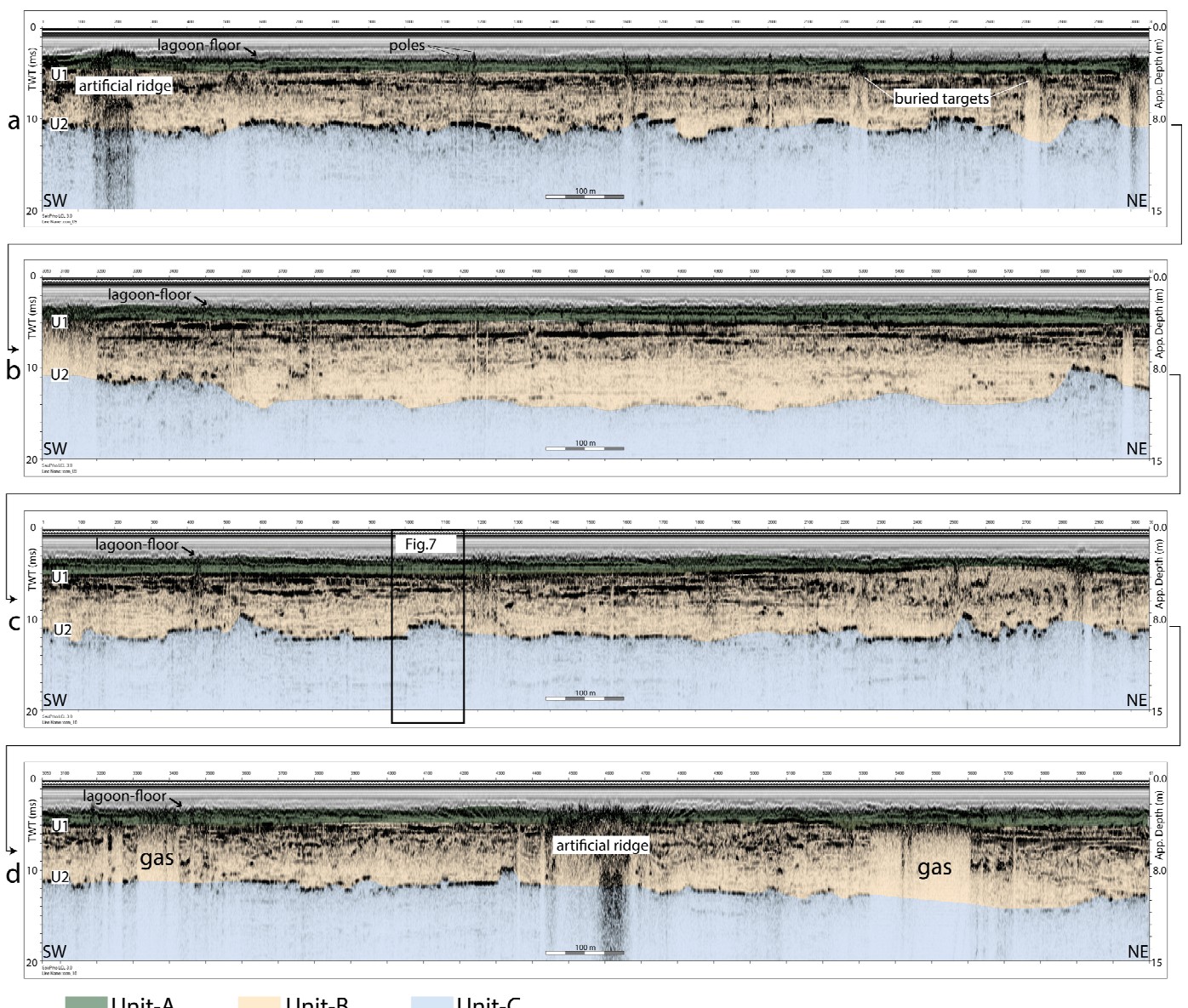

**Figure 6.** (**a**–**d**): Seismic reflection profiles showing main seismostratigraphic units and unconformities (U1 and U2). Units-A, -B and -C are indicated as well as major anthropic and natural peculiar reflection patterns (location in Figure 2).

As previously mentioned, the lagoon's system was deeply modified by human activities over the centuries. Consequently, reflection patterns generated by anthropic features were also visualized in the echograms. In Figure 6, over the lagoon-floor, the acoustic images show the remains of numerous vertical wooden poles, historically used to delimit the navigable channels and to fix fishing gears. Likewise, high-reflectivity targets producing a long sequence of multiples were interpreted as submerged embankments or ancient ridges. Clearly visible are several gas-escape structures, such as acoustic plumes, intra-sedimentary plumes/seismic chimneys, acoustic blanking/gas curtains.

Analysis of the acoustic *facies* enabled us to classify the observed units into three different patterns: (1) typical, fine-laminated lagoon deposits with good (up to 15–20 m) penetration of the seismic signal where acoustic imaging is best developed; (2) high-reflectivity substrate, indicating the presence of "hard" material of natural or anthropic origin, characterized by very limited penetration of the seismic signal and the presence of

multiple reflections; (3) gas-bearing sediments, with diffuse presence of "blind" windows and poor seismic penetration.

In sectors showing a "typical" stratigraphic sequence, three depositional units were recognized. Units -A, -B, and -C, from top to bottom (Figure 6), are delimited by two major unconformities U1 and U2, both characterized by large amplitude and high lateral continuity in the entire working area.

Correlation between seismostratigraphic units and regional stratigraphy was obtained using CPTu soundings carried out along seismic reflection profiles (Figure 7) suggesting the presence of a sandy/silty unit (Unit-B) topped by a muddy layer (Unit-A) and underlined by a homogeneous unit made of consolidated clays.

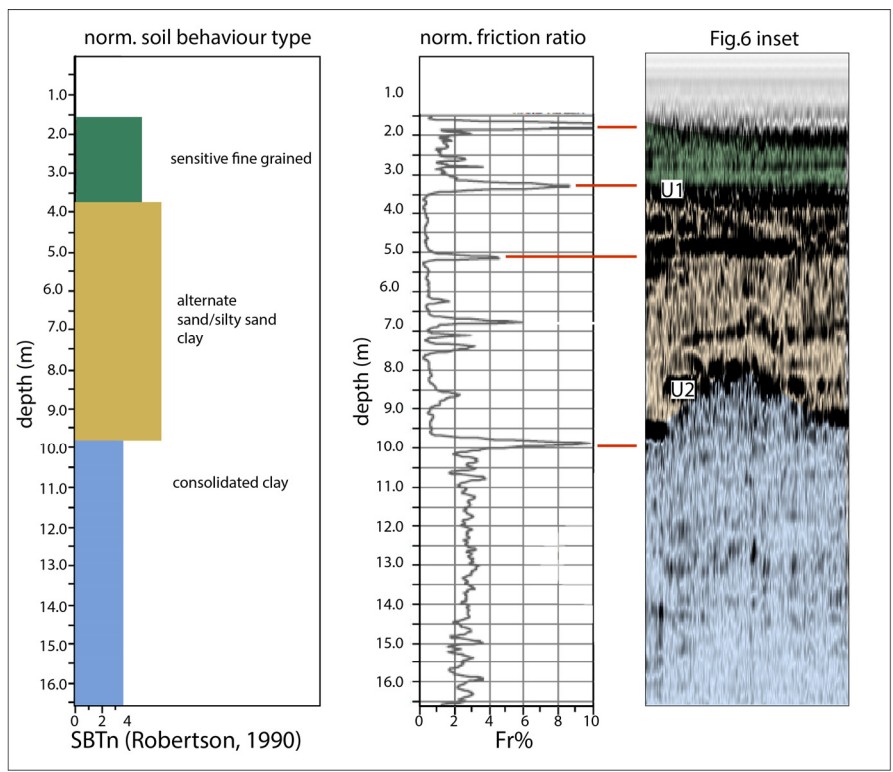

**Figure 7.** Stratigraphic correlations of seismic units and unconformities U1 and U2, resulting from CPTu tests [48]. We note how peaks in the normal friction ratio % fit with strong reflectors, indicating the presence of high acoustic impedance contrasts.

A description of the three recognized seismic units displayed in Figure 6 can be summarized as follows.

Unit-A is the topmost unit, bounded at its base by U1. It shows transparent seismic *facies*, and variable thickness, never reaching up to 3 m. It is made of finely laminated, homogeneous deposits draping the lagoon-floor, possibly bioturbated, and corresponding to the most recent stage of environmental evolution.

Unit-B is bounded by U1 and U2 at its top and bottom, respectively. It is characterized by marked internal reflectors indicating alternate finer- and coarser-grained sediments and complex morphologies, possibly related to the presence of lobes, paleochannels, and sand ridges. Considering geological data obtained close to the site [36,37] and the CPTu soundings (Figure 7), Unit-B is composed by alternating sand, mud, and silt deposits, with complex internal geometries suggesting a relatively high-energy depositional environment. The thickness of this unit is rather constant (5–6 m), if we exclude the presence of ridges, probably made of coarse-grain deposits, where it appears thicker. While some of these features may be natural, some strong acoustic anomalies may be due to the presence of artificial embankments, which show a distinctive reflection pattern. We correlate the Unit-B

deposits to the prograding stacking patterns of the Po River delta dating back to the late Holocene [36,37].

Unit-C is bounded at its top by U2 and is composed of very homogeneous and transparent deposits, not showing internal layering. We do not observe any base, and this might imply that the unit could be relatively thick, although no data could be retrieved below 7–8 m. According to the CPTu tests (Figure 7), the lower Unit-C may consist of fine-grained deposits, and the absence of internal reflectors suggests that it is made of homogeneous consolidated mud, probably of marine origin. In absence of direct samplings, we interpret this unit as being composed by prodelta clays deposited after the maximum Holocene transgression, i.e., around 6 ka [24].

The proposed stratigraphic reconstruction, although representing only the shallower part of the Quaternary sequence, agrees with analysis carried out north of the lagoon, as well as with seismic data from the inner shelf close to the study area [36,37]. It suggests a complex geological history, controlled mainly by eustatic sea-level variations, local subsidence, and sediment supplied by rivers, the latter also strongly influenced by anthropic activity. However, due to the discontinuous and anthropically controlled connection with the eustatic system, particularly in the late Holocene, it would be difficult correlating more precisely our seismostratigraphic observation to fine stratigraphic observations in absence of sediment cores from the lagoons.

### 3.2. Present-Day Setting: Bottom Morphology and Substrate Types

3.2.1. Bathymetric Survey

The bathymetry of the lagoon (Figure 8) is extremely flat, if we exclude artificial embankments and channels created and abandoned during the centuries, as well as natural ridges probably pre-dating the Holocene transgression. Data reveal that the average depth is about 0.6 m, while maximum depths never reach the −2 m isobath. Minimum depths occur irregularly along the margins of the system and in some spots inside the lagoon where they may have an anthropic origin.

However, the signature of natural reliefs, such as sand bars or paleo-dunes, and their modifications, were not completely obliterated by the mud draping of the present-day lagoon, and are still locally visible. A notable example is the Boscoforte Peninsula, which cuts the system into two parts and partially develops under the water with ancillary ridges reaching the northern shore (Figure 8). This peculiar feature was a continuous emerged barrier over which an ancient Roman road (Via Popilia-II century B.C.E) connecting Ravenna to Adria [56] was built taking advantage of existing topography. The Boscoforte peninsula creates a physical separation between eastern and western sectors of the lagoon, which developed different depocenters.

Another interesting feature is a mound-shaped topographic high centered in the southeastern sector, which is probably of anthropic origin as verified from a time series of high-resolution satellite images. Ancient maps, particularly two detailed ones produced in 1580 C.E. and 1658 C.E. (available in https://www.ferraradeltapo-unesco.it/mediateca/#cartografia, consulted 23 July 2020) suggest that this and other minor features, still be recognized around the lagoon, may have been built over ancient pre-existing ridges.

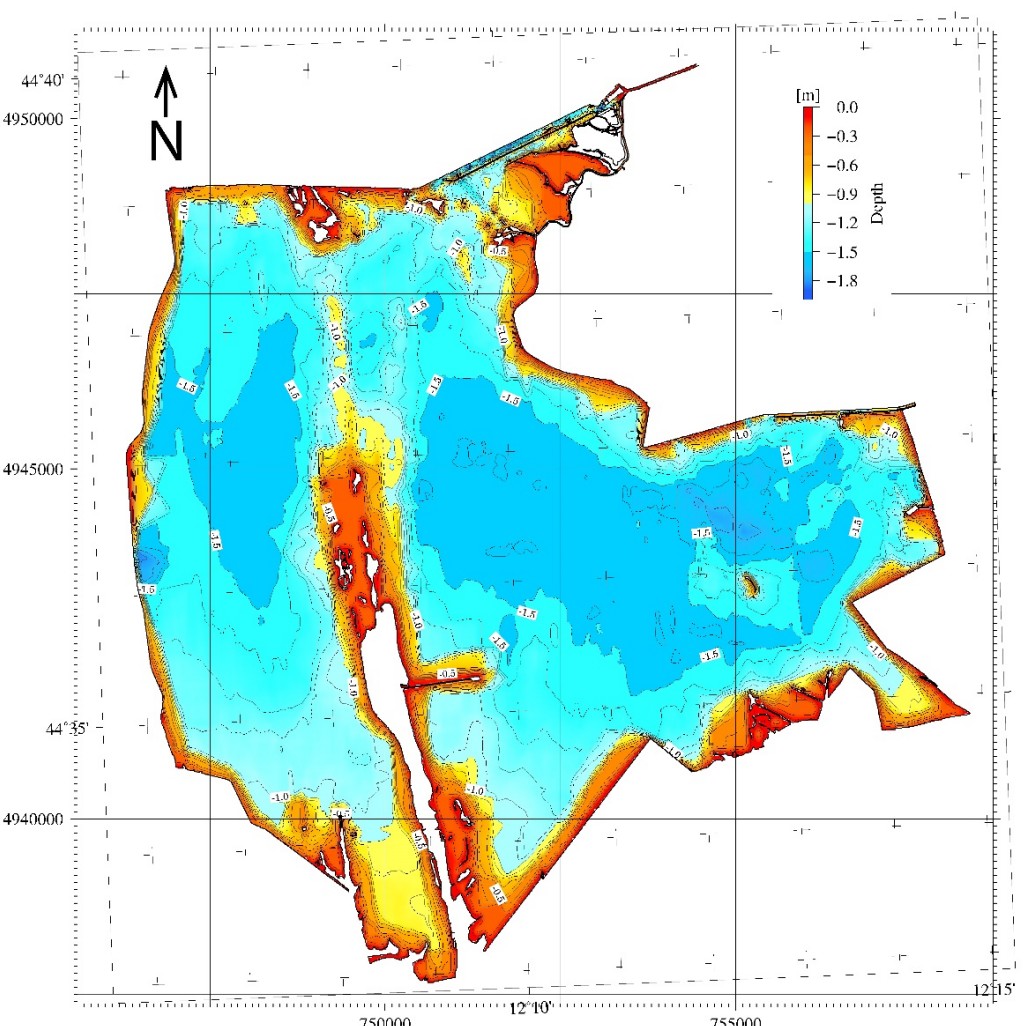

**Figure 8.** Morphobathymetric map of the Valli di Comacchio lagoons compiled using the newly collected dataset (Isolines each 0.5 m; Projection UTM 32N).

### 3.2.2. Bottom Reflectivity and Substrate Variability

Bottom reflectivity data obtained from the echograms were used to compile the map displayed in Figure 9.

The robust correlation between lagoon-floor reflectivity and mean grain size displayed in Figure 3 was used to obtain an estimate of sediment type distribution within the lagoon.

West of the Boscoforte peninsula, we note lower reflectivity patterns relative to those in the eastern sector (Figure 9), suggesting differences in the mean grain size of the sediments at the water-sediment interface (see Section 2.2.2). This could be explained by the coarser aeolian sediment inputs from the coast towards the East, with the western part partially sheltered by the morphological barrier of the Boscoforte Peninsula. The observed bimodal reflectivity pattern may reflect this occurrence. The Boscoforte Peninsula, which shows a clear underwater signature in the bathymetry (Figure 8), but not in the reflectivity map (Figure 9) is most probably a "fossil" ridge system that delineated the easternmost coastline limit at the Etruscans times [23]. Presently draped by fine-grained lagoon sediments, it is aligned with an ancient coastline that delimits two different phases of Po River prodelta sedimentation (Etruscan phase to the west and Roman phase to the east), fully developed around the fifth century BCE [57].

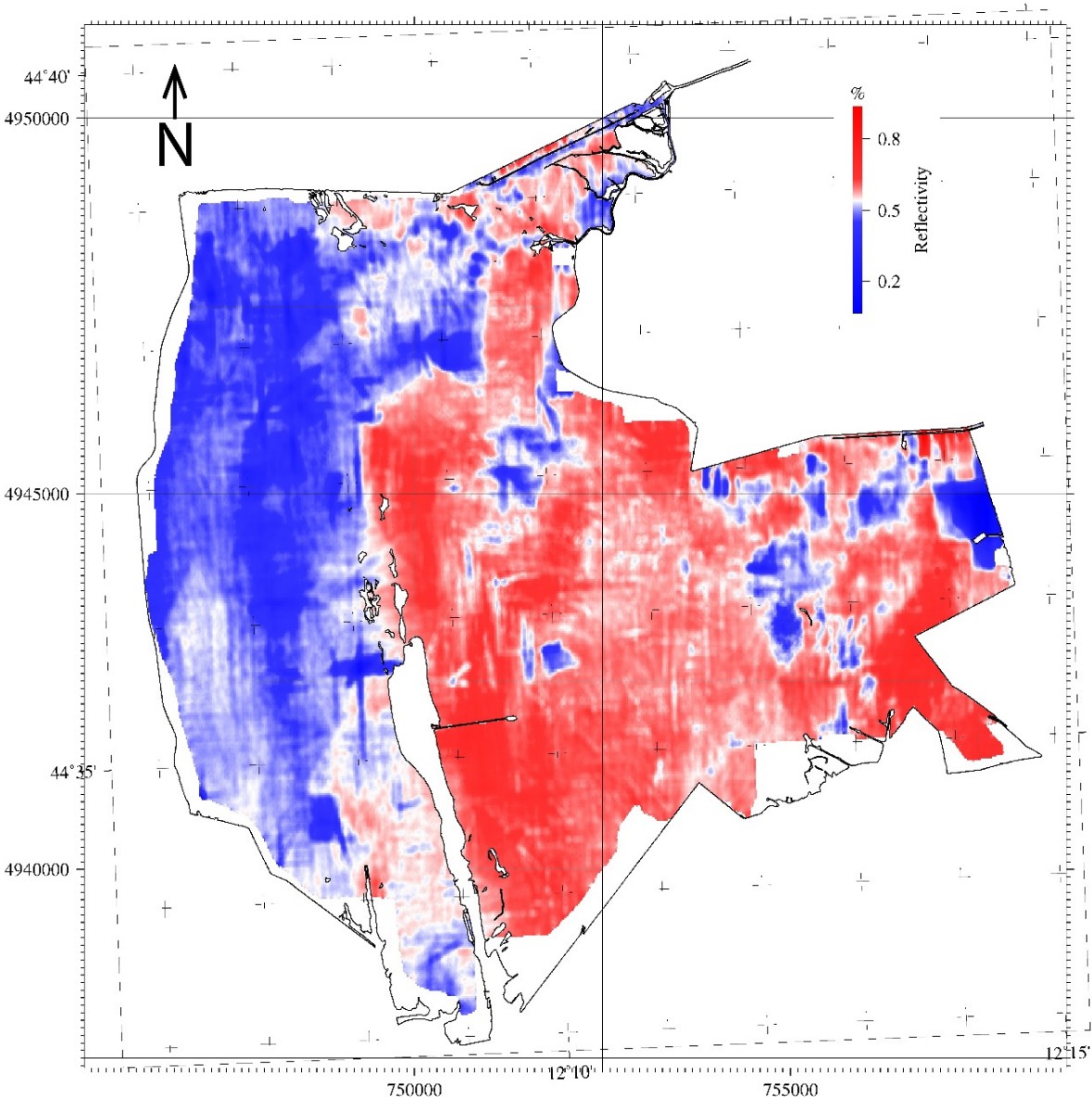

**Figure 9.** Normalized lagoon-floor reflectivity map of the Valli di Comacchio. Reflectivity is displayed using a color palette ranging from red (high values) to blue (low values). Projection UTM 32N.

In Figure 9, we also note a "patchy" reflectivity pattern, with alternating close-spaced highs and lows, particularly evident in the easternmost sector of the lagoon. Although this occurrence could be related to surficial scatter noise randomly interpolated by gridding, it finds a good fitting with feature observed in the time-slice map of Figure 10, less sensitive to such kind of artifacts. We thus may assume that the patchy pattern in the eastern sector is real and corresponds either to the presence of fluid venues (groundwaters or marine-water intrusions) or to damping of excavated material. The latter occurrence is supported by evidence that this sector of the lagoons has been affected by intensive reworking of sediments due to the creation of artificial dams.

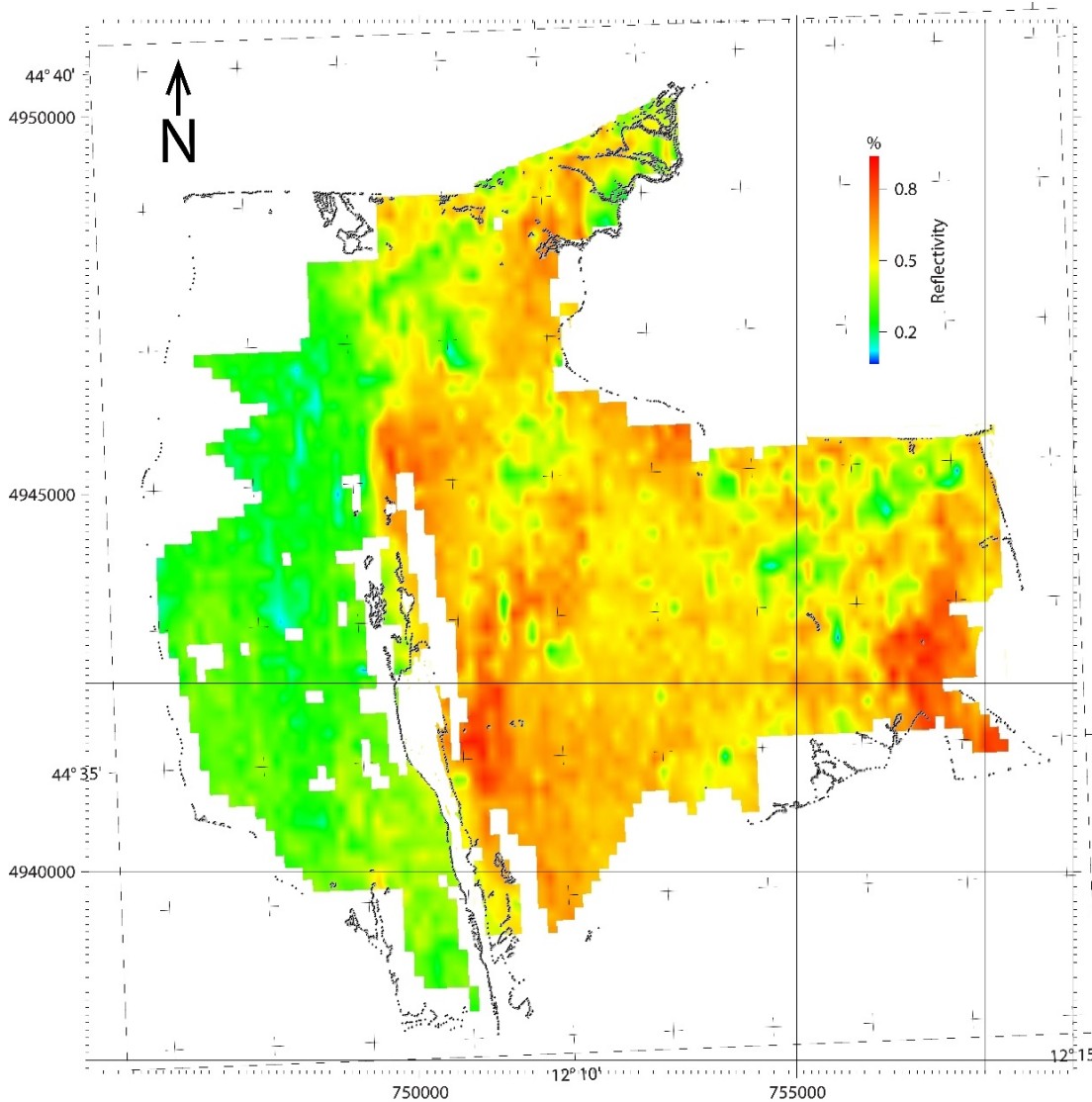

**Figure 10.** Time slice map of the lagoon bottom obtained by integrating first 0.5 m of sediment below the lagoon-floor.

To further test the robustness of the reflectivity analysis, we compiled a *Time-Slice* map, which integrates reflectivity strength of the first 1 ms below the lagoon-floor, considered representative of the latest stage in its evolution recorded by Unit-A (Figure 10). We note a substantial agreement between patterns displayed by the two maps. This might suggest that our analysis is robust, and photographs a relatively stable setting through time, at least for the uppermost sedimentary sequence, when the area reached its present environmental setting. Such occurrence suggests that natural variables (wind transport and local shelters) rather than anthropic activity, is the cause of this bimodal sediment distribution.

### 3.2.3. Topography

A grid of topographic data based on a Lidar survey around the Valli di Comacchio lagoons (https://ambiente.regione.emilia-romagna.it/en/geologia/geology, 1 February 2019) highlights the morphological setting of the study area (Figure 11). We observe the presence of natural sandy ridges along the coast exceeding 5 m in height (red areas in Figure 11), while the inland to the west of the lagoons lays some meters below the mean sea level (blue areas in Figure 11).

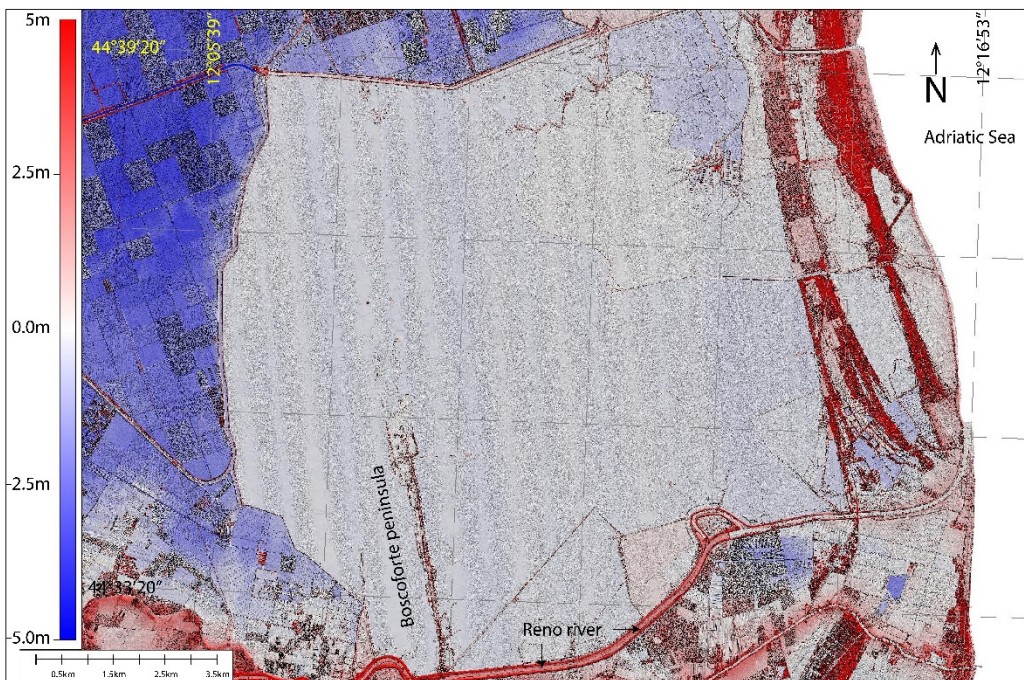

**Figure 11.** Lidar-based topographic model of the inlands surrounding the Valli di Comacchio lagoons. Red colored areas are above the average sea level, while blue color marks areas below.

We note that during centuries the entire system has been artificially divided into several lagoons, taking advantage of the existing barriers and beach ridges (Figure 12). They were considered independent flooded units with different degrees of confinement and managed autonomously. To the west of the Boscoforte Peninsula, the sector composed of sediments with lower reflectivity, probably representing finer-grained lithologies, corresponds to the Valli (or Fossa) di Porto, as indicated by historical maps since the XVI century. Immediately to the east of Boscoforte, we have the ancient Campo della Vacca and Campo da Po (as described in a 1658 map) or Valle Magnavacca, in the nineteen-century cartography (Figure 12), which may also explain this actual difference in local sedimentation.

We note that the present-day inundated surface of the Valli di Comacchio lagoons refers to the area limited by the Po di Volano (the southernmost Po River branch) and the Reno River, to the north and south, respectively, in the frame of the Po River delta territory. Comparing Figures 1 and 12 maps gives us an overview of how deeply the environment was modified in areas below the average sea level at a century time-scale mostly for agriculture development, and how this condition could be considered well balanced, since most of the sediments supplied by the river floods have been subtracted to the system. Da Lio and Tosi [58] report that 65% of Po delta marsh areas have been lost, since the end of the Middle Age through reclamation. This implies that the system can hardly be considered to be in sedimentological equilibrium. From the ecological point of view, it should be stressed that the asymmetrical distribution of bottom types may have implications over the biota, since fine-grained sediments tend to favor accumulation of organic matter [59,60]. Additionally, the limited water circulation results in a higher confinement of the system (lower exchange rates) and may induce anoxic and/or hypoxic conditions [61].

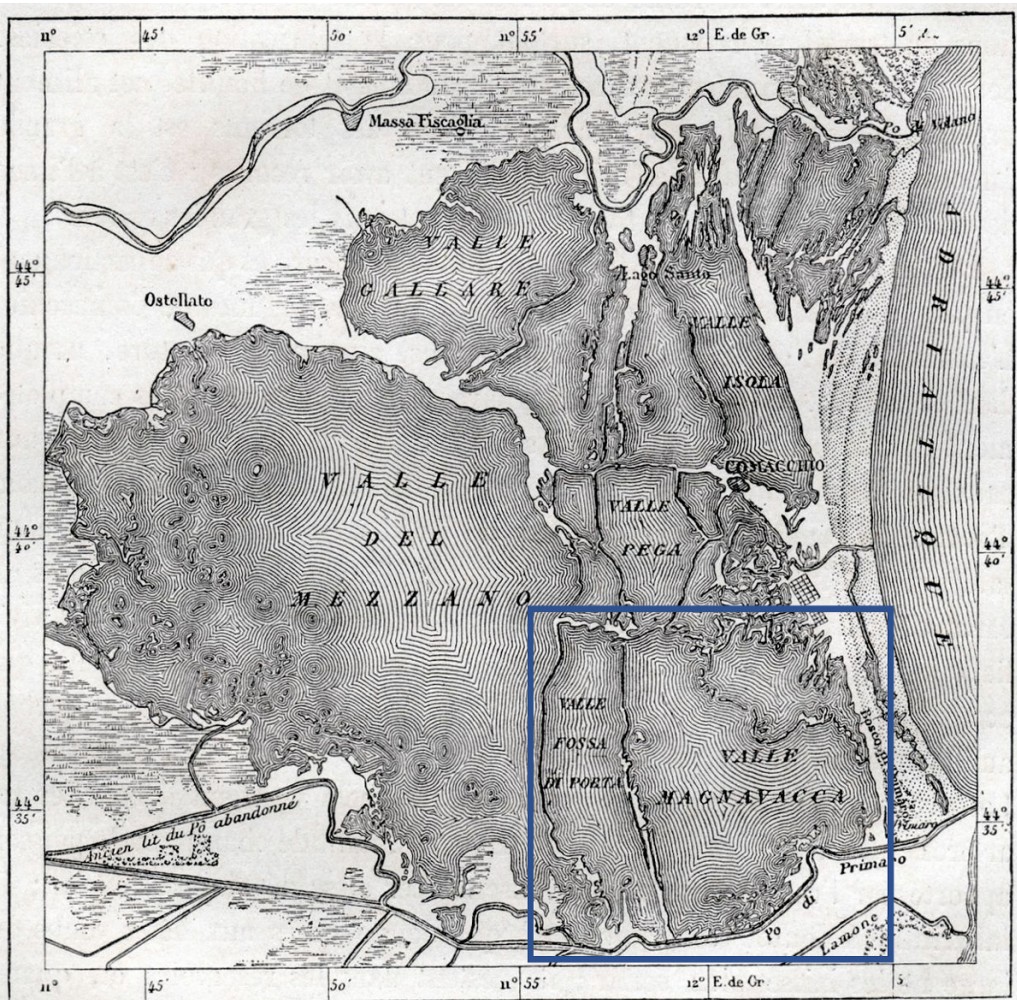

**Figure 12.** The Valli di Comacchio area and its subdivisions represented in a map of 1875 [62]. Note the extensive paludal areas existing to the north and west of the actual system (highlighted in the blue box) that were drained for agriculture development.

Although our approach of correlating reflectivity to the distribution of bottom sediments prevents generalizations, it suggests the presence of variable substrates in the lagoons. In fact, analysis of bottom types based on linear variations between two end-members allowed translating reflectivity images into geological features. This can be particularly useful in those environments where "low-energy" and "high-energy" depositional processes co-exist or are superimposed, or "fossil" vs. "active" features should be recognized.

### 3.3. Hydrodynamics

Hydrodynamics of the lagoon is mostly driven by wind stresses on a free surface and by tidal oscillations, whereas the effects related to wind-generated waves play a secondary role. A recent study [38] simulated the hydrodynamics of the lagoon using a shallow-water equations model based on a Godunov-type high-order numerical scheme combined with unstructured grids.

Wind resulted to be the main factor driving water circulation in the whole lagoon, while tidal waves at the gates affect considerably water speed only within a few hundred meters from the inlets, whereas the siphons effects on the hydrodynamic field are negligible, as a consequence of the low flow rate. Due to wind-induced circulation, the water surface gradient is positive in the wind direction, and it is related to wind intensity. The maximum

simulated difference in water level at opposite borders of the lagoon is about 0.22 m. The geometry of the water body leads to a complex streamline pattern. The water flows in the same direction as the wind in the peripheral and shallower zones, while currents with opposite directions are generated in the deeper central zones. Even in case of extreme wind event (wind speed 11.7 m/s; direction NE) the averaged water velocity is limited to 0.05 m/s, and it reaches maximum values near 0.3 m/s close to the boundaries or in the narrow channels between borders and islands. This leads to a bimodal circulation pattern East and West of the Boscoforte Peninsula, and could explain the observed differences in sedimentation.

### 3.4. Decadal Changes in the Lagoon-Floor Morphology

Post-2016 subsidence rates in the Comacchio lagoons surrounding (Figure 5) agree with estimates performed by other authors [63] and with those obtained by leveling campaigns before 1950 [55]. These data confirm a recent reduction in the subsidence rates trends probably due to more careful environmental policies. Subsidence rates of about 9 mm/y (the higher in the study area) observed at the COCL station (Figure 5) are in line with pre-1950 rates and could be related to the natural trend. It should be pointed out, however, that the GNSS monitoring can provide information limited to the site where the GNSS antenna is located. The lack of geodetic monitoring of the lagoon-floor did not allow for detailed reconstructions of land subsidence rates in the entire study area. This problem could be partially solved by repeated bathymetric surveys.

A comparison between the new bathymetric map (Figure 8) with a previous survey performed in 1992 allowed us to identify changes in local bathymetry of the lagoon at a decadal time scale. A map representing residuals of the two bathymetric datasets is shown in Figure 13. Although the two surveys were carried out with different instruments and associated errors, not allowing for absolute determinations in this research we assumed that those differences are comparable for different sectors of the lagoons. Under this assumption, and neglecting the effects of regional subsidence, we could use Figure 13 map to analyze local changes in different sectors. The entire lagoon shows an overall stable subsiding tendency if we exclude the easternmost sector, where residuals reach up to −50 cm (Figure 13). Here, the Dosso degli Angeli gas reservoir has been active since 1971, with a strong decrease from 1998 to 2004 and a suspension in 2004 [64]. In 2012, the oil company managing the reservoir planned to complete the exploitation of the residual reserves over the period from 2013 to 2023. A maximum of 2.8 cm of subsidence during this latter time interval was predicted using an elasto-plastic Finite Elements Model [64]. According to this work, land subsidence due to the residual gas production from the reservoir will not affect the environment, hydraulic safety, and infrastructures of the Comacchio lagoons and the nearby coast.

The actual geomorphological configuration of the Valli di Comacchio lagoons is strictly related to the interaction of marine and fluvio-deltaic processes, particularly linked to Po River delta evolution. On a millennial scale, the sediments deposited by rivers towards the sea attempt to counterbalance the natural basin subsidence, helping to sculpture the local coast over time [26].

Over the last 2500 years the lowering of ground elevation, in comparison to the Adriatic mean sea level, occurred at a rate of about 2–3 mm/year [65], in agreement with our geodetic data analysis. Our data suggest that the effects of enhanced subsidence due to water and gas pumping have been limited to the southeastern sectors of the lagoon system.

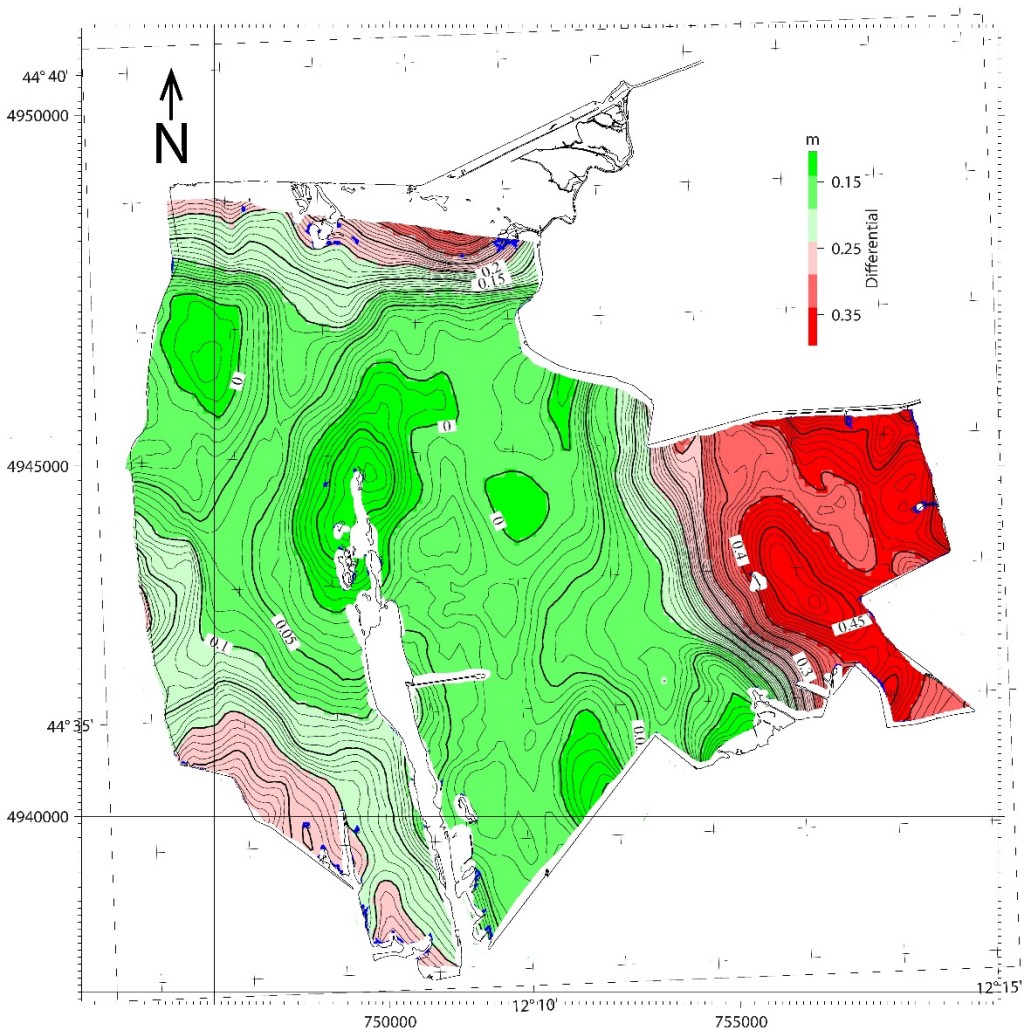

**Figure 13.** Residual bathymetric map obtained subtracting data collected at about 20 years of time-interval (old-new). Green color indicates stable (or regionally subsiding) areas while negative values (towards red) faster differentially subsiding sectors (Projection UTM 32N). Note maximum differential subsidence in the eastern sector.

### 3.5. Morpho-Stratigraphic Synthesis and Integrated Analysis

As pointed out by previous studies in Valli di Comacchio area, most of them based on the analysis of sediment cores collected from boreholes, the high-stand deposits post-LGM are characterized by progradational stacking patterns of prodelta clays, delta front sands (beach-ridge), and delta plain clays and peats, documenting the development of the early, wave-dominated Po delta since late Holocene. In particular, the role of barrier sand ridges in controlling the local sedimentary evolution and human occupation is compatible with regional-scale observations and historic cartography. These extensive barriers and ridge systems (Figure 14) present features progressively younger from west to east and have been described in regional geomorphological analysis [57,66]. These authors, based only on onshore fieldwork, argued that some of the ridges are buried and others exposed. Our findings, and particularly the arcuate high-reflectivity features observed in Figure 10 map, indicate that they may be also submerged, mostly due to the subsidence processes, and their signature is detected by subbottom reflectivity analyses. It is interesting that from the eight ridge systems identified in [57], six are correlated to historical post-Etruscan times (about 500 years B.C.E.), indicating a recent development.

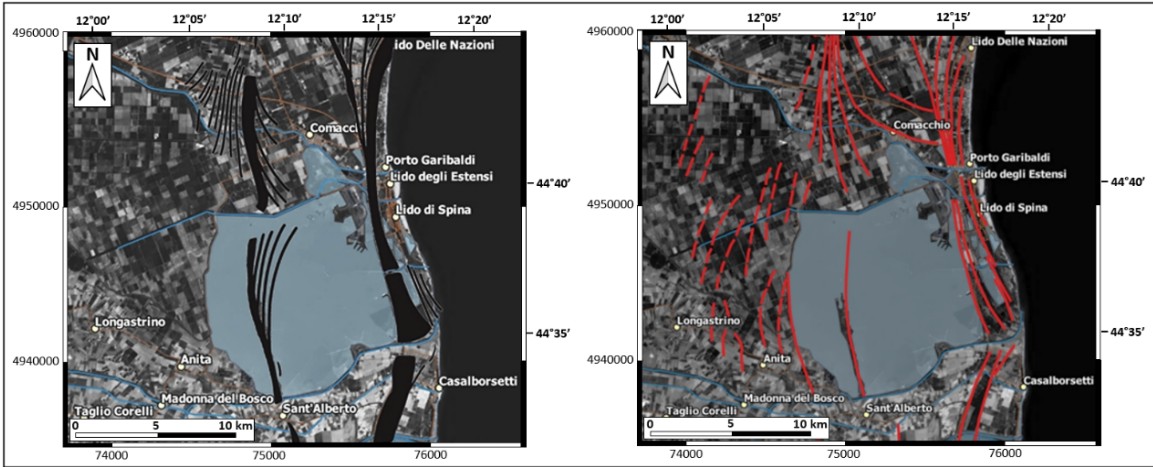

**Figure 14.** Distribution of most prominent barriers and ridges in the Valli di Comacchio area, as described in [57], left, and [66], right map.

Besides their importance in reconstructing the recent geological evolution of the area, ridges also seem to have provided the basis for the actual underwater morphological setting of the lagoon, as well as to network development and economic activities that promoted the occupation of the area during the last millennia. The integrated analysis of lagoon morphobathymetry and its recent changes, sub-bottom structures, and reflectivity of the sediment-water interface provided a comprehensive picture of the depositional setting of the area, in agreement with geomorphological analyses carried out onshore. Seismic reflection images suggest that most of these features are a legacy of the post-Glacial geological evolution of the lagoons recorded in the sedimentary sequence.

The present-day morphological setting, in turn, allows identification of spatial heterogeneities, also related to the geological evolution and historical use. A partially submerged Holocene sand barrier, the Boscoforte peninsula, divides the system into two major sectors and seems to have caused the development of independent basins with individual depocenters and apparently distinct sediment class predominance. Although the Valli di Comacchio lagoons are presently characterized by low-energy depositional processes marked by fine-grained lithologies, the reflectivity and time-slice maps of Figures 9 and 10, which indirectly express the physical properties of the sediments, show composite patterns reflecting local evolution, natural processes, and intensive anthropic transformations. In this context, the differential subsidence at a decadal time scale evidenced by the map in Figure 13 confirms that man-induced alterations are still taking place.

## 4. Conclusions

Although in modern cartography the Valli di Comacchio lagoons are represented as a continuous water body, a geological/geophysical investigation of their substrate shows important vertical and lateral heterogeneities, possibly the legacy of a complex geological evolution, and/or man-induced modifications during several centuries of intensive anthropic pressures. High-resolution seismic reflection profiles and bathymetric data enabled us to describe the late Holocene evolution of the lagoons, as well as the present-day depositional setting and decadal morphological changes. All evaluated proxies confirm that the natural environment created by the Holocene transgression of the Adriatic Sea at the mouth of the Po River has been significantly altered by human activities at different scales. Seismostratigraphic analysis suggests that the Valli di Comacchio sedimentary sequence is made of: -a topmost fine-grained unit with thickness ranging between 1 and 2 m deposited in the low-energy environment of a lagoon confined by damming and river diversions; -a thicker (from 5 to 10 m) composite unit made of alternate muddy and sandy layers of the Po River and tributaries when they were not artificially confined to their

present positions; -a basal unit made of homogeneous consolidated mud below an irregular erosional unconformity.

Comparison of morphological and reflectivity maps gave insights on the physical status of the environment and on the effects of natural and anthropically induced morphological changes, such as the formation of ridges and depressions. Such evidence agrees with independent historical observation and field data onshore, which provide important clues on man-induced alterations.

Comparison between newly collected bathymetry with previous data revealed that the eastern sector of the lagoon underwent enhanced differential subsidence (exceeding 2.5 cm/y over regional datum) which probably caused modifications in hydrology, including water exchanges with the sea.

We observe that the Valli di Comacchio lagoon system underwent a massive anthropic impact since its formation after the Last Glacial Maximum, reflecting demographic, industrial and agricultural development of the region. This long-term experience suggests that when a natural system is deeply modified to fulfill the ever-changing needing of the population, which depends on its resources, all variables should be accounted for, including possible environmental crises generated by non-sustainable development models. To avoid such crises, the knowledge, at the best possible level of accuracy of the geological history of such environments, constitutes a first essential step.

Beyond these general considerations, we stress that submerged continental and transitional areas are effective environmental archives, which could be deciphered only combining multiple sources of data to obtain a comprehensive picture supporting ecological studies and management plans.

**Author Contributions:** Conceptualization, L.G., J.B. and A.P.; methodology, L.G., J.B., G.S., L.S., F.D.B., L.S. and N.C.; formal analysis, J.B., L.G., F.M., N.C., S.D. and A.P.; investigation, F.D.B., L.G., G.S. and S.D.; resources, L.G. and G.S.; data curation, L.G., J.B. and G.S.; writing—original draft preparation, L.G. and J.B.; writing—review and editing; visualization, L.G., J.B., F.D.B. and N.C.; supervision, L.G.; project administration, L.G. and J.B.; funding acquisition, L.G., J.B. and L.S. All authors have read and agreed to the published version of the manuscript.

**Funding:** This research was carried out in the frame of the NAIADI (POR-FESR 2014-2020), grant number 001 and Consorzio Ferrara Ricerche. The study was partially supported by the Coordenação de Aperfeiçoamento de Pessoal de Nível Superior-Brazil (CAPES) (International Visiting Grant 8881.337427/2019). Jarbas Bonetti is a Research Fellow of the Brazilian Conselho Nacional de Desenvolvimento Científico e Tecnológico (CNPq) (Grant 307797/2016-3).

**Institutional Review Board Statement:** Not applicable.

**Informed Consent Statement:** Informed consent was obtained from all subjects involved in the study.

**Data Availability Statement:** All data will be made available at http://www.ismar.cnr.it/products/data-sharing?set_language=en&cl=en (10 June 2021).

**Acknowledgments:** The authors acknowledge support from the Consorzio Ferrara Ricerche for this work, which was completed as part of a LIFE E.C. project. Thanks are due to Marco Bondesan for fruitful discussion and suggestions. Technical support of Enrico Dalpasso (ISMAR-Bologna) and Stefano Mosticchio (Coastal Consulting, Bari) has been fundamental to organize and carry out the surveys. Maria Pia Pagliarusco (Parco Delta del Po) kindly helped in many phases of the project. We also thank the following Institutions: ASI, FOGER (Fondazione dei Geometri e Geometri Laureati dell'Emilia Romagna), LEICA-Italpos, Regione Veneto, RING-INGV, Università degli Studi di Bologna, which have kindly made available GNSS recordings data. This work was carried out in the frame of the NAIADI (POR-FESR 2014–2020) project, which included a citizen science activity carried out in cooperation with Liceo Scientifico Righi (Bologna) Stefano Mazza (Università Cattolica di Brescia), Cristiana Natali and Giovanni Brizzi (Università di Bologna) https://www.youtube.com/watch?v=4q67SYJsmTk, accessed on 14 February 2022.

**Conflicts of Interest:** The authors declare no conflict of interest.

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
