# Peer review of "Anatomy of Anthropically Controlled Natural Lagoons through Geophysical, Geological, and Remote Sensing Observations: The Valli Di Comacchio (NE Italy) Case Study"

_remotesensing, doi:10.3390/rs14040987_

Round 1

Reviewer 1 Report

The manuscript is well written. There are certain space for the authors to sharp the presentation. I have the following comments:

(1) The caption---Anatomy of anthropically controlled natural lagoons through 2 geophysical, historical, and remote sensing observations: the 3 Valli di Comacchio (NE Italy) case study

It sounds add to parallel "geophysical, historical, and remote sensing ". You may do not need the word "historical". The term "anthropically" somehow implies that "historical data" will be used. You did use historical data/information to interpret your geophysical and remote sending data.

2. The "Introduction" section is too long and disattractive. You may focus on the scientific components that you will address in this manuscript.

3. Lines 78-79

The focus of this paper is the post-Glacial geological history of the Valli di Comacchio 78 coastal lagoons (Figure 1), in the Po river delta (NE Italy)

It seems not true.  You may redefine or reorganize the structure of the manuscript

4. Your Table 2, about GNSS-derived site velocities

Your tale w suggests that the site velocities obtained from the long-hisotory (> 9 year) and short history (4 years) datasets are the same. The only difference is the uncertainty. This is expected. The agreement between the  long-term and short-term velocities suggests that the position time series follow a nice linear trend. I suggest you illustrate all position time series (only vertical) and mark velocities on the plots. You do not need Table 2.  It does not provide so much useful information.

About the uncertainty----Uncertainties were estimated using the method pro-302 posed in [36]. I did not find this reference. But it seems not a right reference for this statement. In fact, uncertainty is a very important issue of site velocities at a few millimeters per year level. Google "GNSS-derived site velocity, uncertainty", you may find many recent articles.

Your reference 36---may wrong? Bernardi, D., Caleffi, V., Gasperini, L., Schippa, L., Valiani, A. A Study of the Hydrodynamics of the Coastal Lagoon “Valli di 766 Comacchio”. In: 3rd Int. Symp. on Shallow Flows, Iowa City, USA, June 4 2012, 6, 2012.

5. You mentioned "submergence, which is a combination of regional coastal subsidence and sea-level rise.  You may cite sea-level study in this area and give a number about average sea-level rise(?) rate (e.g., past 100 years) in this area, which could be slightly different with the global average (2.0 mm/year?).  Anyway, sea-level rise is important and you need to count this in your analysis.

Author Response

The manuscript is well written. There are certain space for the authors to sharp the presentation. I have the following comments:

(1) The caption---Anatomy of anthropically controlled natural lagoons through 2 geophysical, historical, and remote sensing observations: the 3 Valli di Comacchio (NE Italy) case study

It sounds add to parallel "geophysical, historical, and remote sensing ". You may do not need the word "historical". The term "anthropically" somehow implies that "historical data" will be used. You did use historical data/information to interpret your geophysical and remote sending data.

Observation correct and very appropriate. We modified the title according to the referee’s suggestions. The new title is “Anatomy of anthropically controlled natural lagoons through-geophysical, geological, and remote sensing observations: the Valli di Comacchio (NE Italy) case study”

  1. The "Introduction" section is too long and disattractive. You may focus on the scientific components that you will address in this manuscript.

We significantly shortened the “Introduction” section leaving only those parts discussed in the data analysis and relevant to the focus of the paper. We originally added this other information because very important in the perspective of an anthropic control of the natural environments. We acknowledge, however, that it was too long and too distant from the main topics.

  1. Lines 78-79

The focus of this paper is the post-Glacial geological history of the Valli di Comacchio 78 coastal lagoons (Figure 1), in the Po river delta (NE Italy)

It seems not true.  You may redefine or reorganize the structure of the manuscript

Yes, the referee is right. In fact we are not dealing the Po river delta sensu stricto, but more properly with the “Po river delta region”. We modified the text accordingly. 

  1. Your Table 2, about GNSS-derived site velocities

Your tale w suggests that the site velocities obtained from the long-hisotory (> 9 year) and short history (4 years) datasets are the same. The only difference is the uncertainty. This is expected. The agreement between the  long-term and short-term velocities suggests that the position time series follow a nice linear trend. I suggest you illustrate all position time series (only vertical) and mark velocities on the plots. You do not need Table 2.  It does not provide so much useful information.

Reply: as suggested to the reviewer we have removed  table 2 and added a new figure. The velocities values are reported on the new plots of the vertical component time series. The agreement between the relatively long term and short-term velocities is not an obviously results. Several paper show as human activities can modify the rate of the GNSS sites or other natural processes, as for example, seismic events of seasonal signals with a relatively long period (5, 6, 7 years) can modify the linear trend of the daily time series obtained processing GNSS data.

About the uncertainty----Uncertainties were estimated using the method pro-302 posed in [36]. I did not find this reference. But it seems not a right reference for this statement. In fact, uncertainty is a very important issue of site velocities at a few millimeters per year level. Google "GNSS-derived site velocity, uncertainty", you may find many recent articles.

Reply:  thanks to the reviewer, there is an error in the reference. The corrected references are [34],[52], [53] and [54]. There are reported on the text.

Your reference 36---may wrong? Bernardi, D., Caleffi, V., Gasperini, L., Schippa, L., Valiani, A. A Study of the Hydrodynamics of the Coastal Lagoon “Valli di 766 Comacchio”. In: 3rd Int. Symp. on Shallow Flows, Iowa City, USA, June 4 2012, 6, 2012.

The reference is correct. In one case it was included in the wrong place. We corrected it.

  1. You mentioned "submergence, which is a combination of regional coastal subsidence and sea-level rise.  You may cite sea-level study in this area and give a number about average sea-level rise(?) rate (e.g., past 100 years) in this area, which could be slightly different with the global average (2.0 mm/year?).  Anyway, sea-level rise is important and you need to count this in your analysis.

Yes, we added a discussion on this point. In particular, we cited the work by Amorosi et al, 2017, which discuss relative sealeve changes in the region, to the "Introduction" section, and added the following sentence in the 3. section: "However, due to the discontinuous and anthropically controlled connection with the eustatic system, particularly in the late Holocene, it would be difficult correlating more precisely our seismostratigraphic observation to fine stratigraphic observations in absence of sediment cores from the lagoons”

Reviewer 2 Report

The Introduction section is large, covering different environmental and historic aspects of the Lagoon. However, the main purpose of this study, research goals, or questions are not well described. Please clarify them. In addition, the manuscript needs professional English proofreading.

According to RS Instruction for Authors, it is necessary to include scale, north arrow, geographic coordinates, and legend in all figures representing maps. Please supplement such where necessary.

Specific comments:

line 66-68. I recommend adding references from the other environments, like, e.g.:

  • 10.1002/arp.1823

line 102. Reference needed. Consider:

  • 10.1002/gj.2934
  • 10.1111/sed.12621

line 193: Provide more technical details (including e.g., manufacturer) of the used sub-bottom profiler system.

line 204: better explain, what do you mean by "soupy".

line 218: what do you mean by echosounder sweep?

line 250: Explain CPT abbreviation

Figure 6 / line 427. Clarify what is presented on the map. If the map represents the percentage occurrence of the sand, do not name it acoustic reflectivity. Instead, in the methods section, you should explain how you determined the percentage occurrence of the sand.

line 425. Reference needed

line 440. How does such a "patchy" reflectivity pattern come out from the real acoustic backscatter and how from interpolation between ship track lines? A discussion on this topic is needed.

line 461: the link is not working. By the way, maybe there is topo-bathymetry lidar data available for this area? Such a device would be very efficient in such shallow areas. 

line 494. This approach needs a better explanation in a methodical sense.

line 617. Change the chapter number to 5.

line 634. There is no such a combined map provided in the whole manuscript.

Author Response

The Introduction section is large, covering different environmental and historic aspects of the Lagoon. However, the main purpose of this study, research goals, or questions are not well described. Please clarify them. In addition, the manuscript needs professional English proofreading.

We significantly shortened the “Introduction” section leaving only those parts discussed in the data analysis and relevant to the focus of the paper. We originally added this other information because very important in the perspective of an anthropic control of the natural environments. We acknowledge, however, that it was too long and too distant from the main topics. Finally, we asked for a professional proofreading

According to RS Instruction for Authors, it is necessary to include scale, north arrow, geographic coordinates, and legend in all figures representing maps. Please supplement such where necessary.

OK north arrows, geographic coordinates and legend (where needed or missing) were added

Specific comments:

line 66-68. I recommend adding references from the other environments, like, e.g.:

  • 10.1002/arp.1823

Included

line 102. Reference needed. Consider:

  • 10.1002/gj.2934
  • 10.1111/sed.12621

Included

line 193: Provide more technical details (including e.g., manufacturer) of the used sub-bottom profiler system.

Done

line 204: better explain, what do you mean by "soupy".

We changed “soupy” with “soft”

line 218: what do you mean by echosounder sweep?

Although accepted in technical jargon, “sweep” could reslut ambiguous. We replaced it with “wavelet”

line 250: Explain CPT abbreviation

done

Figure 6 / line 427. Clarify what is presented on the map. If the map represents the percentage occurrence of the sand, do not name it acoustic reflectivity. Instead, in the methods section, you should explain how you determined the percentage occurrence of the sand.

We provide a better description of how we infer sediment distribution from reflectivity, both in the Methods” and “Bottom Reflectivity” sections

line 425. Reference needed

methods chapter referred

line 440. How does such a "patchy" reflectivity pattern come out from the real acoustic backscatter and how from interpolation between ship track lines? A discussion on this topic is needed.

Done:

Although this occurrence could be due, in theory, by scattered superficial noise randomly interpolated by gridding of relatively largely-spaced lines, it finds a good fitting in the time-slice map displayed in Figure 7, less sensitive to such kind of artifacts”.

line 461: the link is not working. By the way, maybe there is topo-bathymetry lidar data available for this area? Such a device would be very efficient in such shallow areas. 

The only bathymetric lidar data available in the region are those relative to the Adriatic Sea coast. In fact. This is probably due to the poor transparency of the lagoons.

line 494. This approach needs a better explanation in a methodical sense.

It is discussed in the methods section

line 617. Change the chapter number to 5.

We made some mistakes in the numbering that were corrected in this new version.

line 634. There is no such a combined map provided in the whole manuscript.

The sentence was ambiguous. We replaced “Combined” with “Comparison of”

Reviewer 3 Report

Dear authors, thank you for the interesting manuscript, I have read it with great pleasure!

The article is well structured, filled with high-quality illustrations, written in scientific language. The involvement of modern scientific equipment and archival data made it possible to obtain interesting results that will find practical application.

Despite the obvious positive aspects, there are some remarks, after correction of which the article can be recommended for publication in the Special Issue: Onshore-Offshore Geophysical and Remote Sensing Techniques for the Study of the Coastal Environment 

2.1 Field Survey

Please show the equipment PSA900 by Datasonics installed on the boat, as well OpenSWAP NAIADI, it will be interesting for the reader, maybe one picture should be dedicated to the equipment. You can also list the technical capabilities of the equipment in the table to understand its applicability in your study.

Seismic data collected at the same time as the bathymetric survey? This can be understood indirectly from the caption to figure 2. It is advisable to indicate in the text or show in the figure with the equipment.

How does the Benthos Chirp III system behave on areas shallower than about 1 m? 

Figure 2 - "Tracks of survey lines" - maybe better "Survey lines". It is desirable to use the same design of cartographic materials, as in Fig. 1 - degree grid and scale bar.

Line 205-206 – "In order to correct data for this effect (soft sediments), we used the method developed in [41]" – Please briefly explain this method. 

2.2.4 Differential bathymetric models

Has the accuracy of the comparison between the old and new bathymetric data been evaluated, given the significant difference in resolution? What is the measurement error?

2.4 Geodetic measurements

Table 1. - since the coordinates on the maps are indicated in the metric system, it is desirable to add the X Y coordinates of the stations in UTM projection. 

3. Results and discussion

Line 292 – Figure 3 instead of Figure 2. Arrange the cartographic material in the same way, as in fig. 1 - degree grid and scale bar. Two degree grids interfere with perception, it is desirable to leave only UTM.

4.1 Recent sedimentary evolution: the shallow subsurface – should be 3.1 !!! The remaining subparagraphs must also begin with 3

Was the data obtained layer-by-layer (Fig. 4) verified in the test sites? If not, what was the basis for the seismic reflection profiles interpretation? 

4.2.1 Bathymetric survey

"Data reveal that the average depth is about 0.6 m" - but Figure 5 show other depths (1.2-1.8, maybe 1.6?), or are you talking about the littoral part only?

Have you tried to find out how subsidence and sedimentation are related? Or, vice versa, to determine by the rate of subsidence, assuming that it is constant and uniform, the thickness of sediments accumulated over 20 years along the entire bottom of the lagoon. 

Author Response

Dear authors, thank you for the interesting manuscript, I have read it with great pleasure!

Thanks you! we are glad you appreciated our work

The article is well structured, filled with high-quality illustrations, written in scientific language. The involvement of modern scientific equipment and archival data made it possible to obtain interesting results that will find practical application.

Despite the obvious positive aspects, there are some remarks, after correction of which the article can be recommended for publication in the Special Issue: Onshore-Offshore Geophysical and Remote Sensing Techniques for the Study of the Coastal Environment 

2.1 Field Survey

Please show the equipment PSA900 by Datasonics installed on the boat, as well OpenSWAP NAIADI, it will be interesting for the reader, maybe one picture should be dedicated to the equipment. You can also list the technical capabilities of the equipment in the table to understand its applicability in your study.

Reply: We do agree with the referee. However, due to other requests, we were forced to add 2 extra figures, and the total number is now 13, which we consider a limit. Through cited papers, and in particular Stanghellini et al., 2020, the readers (and the referee) could find pictures and technical description of the instruments use

Seismic data collected at the same time as the bathymetric survey? This can be understood indirectly from the caption to figure 2. It is advisable to indicate in the text or show in the figure with the equipment.

Yes

How does the Benthos Chirp III system behave on areas shallower than about 1 m? 

Very noisy. In fact, in those cases we were forced to use only data from the 200 kHz echosounder

Figure 2 - "Tracks of survey lines" - maybe better "Survey lines". It is desirable to use the same design of cartographic materials, as in Fig. 1 - degree grid and scale bar.

Corrected

Line 205-206 – "In order to correct data for this effect (soft sediments), we used the method developed in [41]" – Please briefly explain this method. 

It was also required by REF2. We added a more detailed explanation in the methods section

2.2.4 Differential bathymetric models

Has the accuracy of the comparison between the old and new bathymetric data been evaluated, given the significant difference in resolution? What is the measurement error?

While errors can be evaluated for the new survey (and we described in the methods section how they were evaluated) it is impossible for the old survey. Accordingly, we discuss only relative differences between the two surveys and cannot discuss the absolute values.

2.4 Geodetic measurements

Table 1. - since the coordinates on the maps are indicated in the metric system, it is desirable to add the X Y coordinates of the stations in UTM projection. 

Done

  1. Results and discussion

Line 292 – Figure 3 instead of Figure 2. Arrange the cartographic material in the same way, as in fig. 1 - degree grid and scale bar. Two degree grids interfere with perception, it is desirable to leave only UTM.

This is in contrast to what required by the other referees. For such a reason we decided to include, when possible both reference systems

4.1 Recent sedimentary evolution: the shallow subsurface – should be 3.1 !!! The remaining subparagraphs must also begin with 3

Yes, corrected

Was the data obtained layer-by-layer (Fig. 4) verified in the test sites? If not, what was the basis for the seismic reflection profiles interpretation? 

Yes, since we have three units very different one from the other in term of lithology, it was relatively easy and robust correlating seismostratigraphic units to local stratigraphy. This could be considered valid only for the major units, while for a more detailed stratigraphic reconstruction, sediment core analysis would be required.

4.2.1 Bathymetric survey

"Data reveal that the average depth is about 0.6 m" - but Figure 5 show other depths (1.2-1.8, maybe 1.6?), or are you talking about the littoral part only?

Yes, “average depth” could be misleading. Thus we eliminated such non critical information

Have you tried to find out how subsidence and sedimentation are related? Or, vice versa, to determine by the rate of subsidence, assuming that it is constant and uniform, the thickness of sediments accumulated over 20 years along the entire bottom of the lagoon. 

This could have been a very interesting point. However, the stratigraphic resolution of our data was not high enough to focus on that relatively short interval.

Reviewer 4 Report

The manuscript presents an interesting integrated study that highlights the post-Glacial evolution of the Valli di Comacchio coastal lagoons. The paper is well organized and the data and results are properly described. The introduction presents a long description of the study area which should be moved to a specific "Study Area" section.
A weakness of the entire manuscript is the lack of comparison with studies conducted in other similar geological context, especially in Italy (i.e. the Volturno, Sele, Tiber, Arno, Metaponto delta systems - see selected references below). It is fundamental to highlight the novelty of the study approach with respect to the international panorama.
Other specific comments are included in the pdf file.

In my opinion, the manuscript can be accepted with only minor revision.

Kind regards

NOTE:

Aiello, G., Amato, V., Aucelli, P.C.C., Barra, D., Corrado, G., Di Leo, P., Di Lorenzo, H., Jicha, B., Pappone, G., Parisia, R., Petrosino, P., Russo Ermolli, E., Schiattarella, M., 2021. Multiproxy study of cores from the Garigliano plain: an insight into the late quaternary coastal evolution of central-southern Italy. Palaeogeogr. Palaeoclimatol. Palaeoecol. 567, 110298.

Amato, V., Aucelli, P.P.C., Ciampo, G., Cinque, A., Di Donato, V., Pappone, G., Petrosino P., 2013. Relative sea level changes and paleogeographical evolution of the Southern Sele plain (Italy) during the Holocene. Quaternary International, 288, 112-128.

Buffardi, C., Barbato, R., Vigliotti, M., Mandolini, A., Ruberti, D., 2021. The Holocene evolution of the Volturno coastal plain (northern Campania, southern Italy): implications for the understanding of subsidence patterns. Water, 13. https://doi.org/10.3390/xxxxx

Di Rita F, Celant, A Magri D (2010) Holocene environmental instability in the wetland north of the Tiber delta (Rome, Italy): sea-lake-man interactions. Journal of Paleolimnology 44: 51-67. doi: 10.1007/s10933-009-9385-9

Milli S, D'Ambrogi C, Bellotti P, Calderoni G, Carboni MG, Celant A, Di Bella L, Di Rita F, Frezza V, Magri D, Pichezzi RM, Ricci V (2013) The transition from wave-dominated estuary to wave-dominated delta: The Late Quaternary stratigraphic architecture of Tiber River deltaic succession (Italy). Sedimentary Geology 284-285, 159-180. doi:10.1016/j.sedgeo.2012.12.003

Tropeano, M., Cilumbriello, A., Sabato, L., Gallicchio, S., Grippa, A., Longhitano, S.G.,  Spilotro, G., 2013. Surface and subsurface of the Metaponto Coastal Plain (Gulf of Taranto-southern Italy): Present-day- vs LGM-landscape. Geomorphology, 203, 115–131. doi: 10.1016/j.geomorph.2013.07.017

Author Response

Replies referred to the annotated PDF provided by the referee:

As suggested by the referee we moved the description of the study site in a new (1.1) section

What is the evidence for the interpretation of facies? The description of units A, B and C is extremely detailed but based solely on data collected in neighboring areas. Are in situ stratigraphic data available?

Reply: No, unfortunately no direct stratigraphic observation are available for the lagoons. However, we used CPTu test to correlate stratigraphic analyses carried out near our site to seismostratigraphic observations

Round 2

Reviewer 2 Report

My comments were addressed properly. Thank you